# The Development of the Hospitality Sector Facing the Digital Challenge

**DOI:** 10.3390/bs12060192

**Published:** 2022-06-15

**Authors:** André Riani Costa Perinotto, Sávio Machado Araújo, Vicente de Paula Censi Borges, Jakson Renner Rodrigues Soares, Lucília Cardoso, Luís Lima Santos

**Affiliations:** 1Tourism Department, Parnaíba Delta Federal University, Parnaíba 64202-105, Brazil; vpborges@ufpi.edu.br; 2Tourism Department, State University of Ceará, Fortaleza 60714-903, Brazil; savio.machado@sefaz.ce.gov.br; 3Faculty of Tourism, University of A Coruña, 15008 A Coruna, Spain; jakson.soares@udc.gal; 4CiTUR—Centre for Tourism Research, Development and Innovation, Polytechnic of Leiria, 2411-901 Leiria, Portugal; lucyalves.lucilia@gmail.com

**Keywords:** developing hospitality, digital challenge, consumer buying behavior, electronic commerce, conceptual model of acceptance and use of technology

## Abstract

The widespread use of the Internet has changed consumer buying behavior, especially among tourists, considering the intangibility of the tourist product. Although globally the modern tourist is a consumer of the new online market, there is a lack of studies addressing the level of development of the hotel business in relation to online bookings by residents of tourist destinations. Furthermore, this article analyzes the factors that impact the use of e-commerce by these residents in the acquisition of tourist accommodation services. A conceptual model was adopted, using the constructs “social influence” and “price” of the unified theory of acceptance and use of technology 2 (UTAUT2), also including “trust” and “perceived risk”. An electronic questionnaire was used to collect data from a sample of 195 residents of Fortaleza, Ceará, Brazil. Data analysis included descriptive statistics, exploratory factor analysis, Fisher’s exact test, and multiple linear regression. The results show that price and social influence are the most significant constructs associated with booking a hotel online, while trust does not influence the use of the Internet. As practical implications, the findings of this study provide important information for hotel managers, as they allow a better understanding of the profile of respondents who book online, as well as which factors influence online behaviors, contributing to increasing the knowledge of digital platforms in the consumer market and, consequently, the development of the hospitality sector.

## 1. Introduction

In an increasingly globalized world, the Internet impacted advertising, distribution, delivery of products and services, and sales revenues [1,2] due to the speed, ease, and convenience with which information can be accessed. This, in turn, led to significant transformations in the consumer market. In this sense, the use of the Internet grows worldwide every year, and, as corroborated by the International Telecommunication Union [3], by the end of 2019, the number of individuals who use the Internet globally reached around 4.1 billion people (53.6% of the world’s population).

Such a level of utilization [4] allows for the emergence of new businesses, based on innovative ideas, proposals, products, and services made possible by the access and sharing of information available on the Internet [5]. In this context, Jaremen [6] states that the innovations that emerged with information and communications technologies (ICTs) changed the way tourism and hospitality companies conduct their business, given that the purchasing process of tourism services is specifically based on the collection of information available [7].

Furthermore, the Internet emerges as an ideal partner for the tourism and hotel market, since consumers, when first planning their trip and staying in a destination, must make their decision without touching or looking at the “product” [8]. Indeed, Ferreira et al. [9] point out that it is currently possible to plan all stages of a trip online; that is, one can buy airline tickets, book accommodation services, purchase attraction tickets, view maps, rent cars, and seek tips and reviews of tours and restaurants. Therefore, the Internet changed the way tourists contract tourism services. The modern traveler has become the consumer of a new market, known as the online market, or electronic commerce, and user reviews are a new source of information in the hospitality and tourism sector [10].

Additionally, in a survey carried out by Guasti et al. [11], a company specializing in the dissemination of data on e-commerce, in 2017, the online tourism segment (defined herein as the trade of the primary items related to travel, such as airline tickets, hotel reservations, tourism packages, and other services related to online travel agency—OTA) shows the second-highest nominal growth, from BRL 29.8 billion in 2016 to BRL 35.1 billion in 2017, or a 17.8% increase. The survey also reveals that the financial turnaround of this segment accounts for 31.3% of the total resources handled by digital commerce in the Brazilian economy, which totaled BRL 112.19 billion [11]. Therefore, the study of the tourism sector in this research is justified, due to its representation in the revenue and growth of e-commerce in the country.

Still, according to Teixeira et al. [12], the north–east region shows the most significant growth, with a 27% increase and BRL 7 billion in revenues, compared to 2017 (EBIT, 2019). In turn, the Brazilian state of Ceará (CE) is identified as the third-largest online shopping market in the north–east region, during the same period [13].

A survey conducted by the Compre and Confie movement shows that e-commerce revenue in the city of Fortaleza and adjacent, surrounding cities totaled BRL 180.4 million in June 2020 alone, a 163% increase compared to the same month of the previous year (BRL 68.6 million) [14]. However, there is a lack of studies that consider residents in Fortaleza/CE who use e-commerce for tourism purposes, since in most scientific research, Fortaleza is presented as a tourist destination for foreign tourists. Fortaleza is the third most populous city in Brazil, and the most populous in the Brazilian north–east region. To illustrate the demographics and data, we present Fortaleza against Brazil in Table 1.

Given the grounds and considerations presented herein, and realizing that more and more tourists use e-commerce to purchase accommodation services in the scope of the digital experience [17,18], more specifically online hotel booking, a question is formulated, responses to which will contribute to the development of the hotel sector [19]: What factors influence residents of a tourist destination (Fortaleza, Ceará, Brazil) in the use of electronic commerce for purchasing accommodation services?

The susceptibility of the consumer to the environment in which he/she is inserted is perceptible through the observance of social influences, risk perception, price adequacy, trust, and the consumption profile itself. In the destination studied, the use of the internet is also understood as impacting the dimensions of the use of e-commerce for the purchase of lodging services by tourists residing in the city of Fortaleza.

Departing from that, this study aims to analyze the scope of the factors trust, price, perceived risk, and social influence in the use of electronic commerce for the purchase and acquisition of accommodation services and, consequently, the impact on hotel revenues [1], considering the residents of a Brazilian metropolis who are used to conducting business through digital media as the research population.

Moreover, this study contributes to the literature by identifying and analyzing the factors influencing the use of e-commerce for booking accommodations, more specifically among residents of Fortaleza, Ceará, Brazil who wish to travel, and booked their accommodations over the internet. Therefore, the meaning of this research is also fixed, in addition to the contribution already mentioned, to the breadth of the study, which presents data on variables such as habit, hedonic motivation, confidence, performance expectancy, security, privacy, satisfaction, skill, and frequency of use, which, in turn, allows for the conduction of new studies, and are presented as determinant factors in the use of the Internet for tourist purposes [20,21,22,23]. However, so far, no study on the theme addressed here, considering the conjunction of the analysis of the variables described, has been found because, normally, one finds research that use them in isolation, or with application in contexts other than the one proposed here, highlighting a new research perspective.

To proceed, the data collection tool chosen was the questionnaire survey, which was applied via the Internet, distributed via Instagram and WhatsApp. Regarding data analysis, descriptive statistics were adopted, as well as multivariate analysis using Fisher’s exact test, exploratory factorial analysis, and multiple linear regression using the Statistical Package for the Social Sciences (SPSS), version 22.0.

Among the results achieved, it stands out that, unlike price and social influence, the trust dimension did not have a relevant impact on the use of e-commerce for the purchase of hosting services.

For a better understanding, this paper presents a structure that enables the unleashing of logical reasoning about the issues pertinent to the research developed. Thus, the theoretical review was based on the two subjects central to the study, e-commerce and tourism, structured in a systematic and integrative manner, and combining knowledge and information from the empirical and theoretical literature. Next, the conceptual model and the set of hypotheses that support the study were made explicit, in addition to detailing the research methodology, which brings with it the pertinent scientific rigor.

## 2. Electronic Commerce and Tourism

Electronic commerce is a form of commercial transaction for goods and services through the internet, popularly known as e-commerce. In 1998, the WTO adopted a broad definition of electronic commerce: the production, distribution, marketing, sale, or delivery of goods or services by electronic means. The definition coined almost 20 years ago seems to be more suitable to define what is now called “digital economy”, which is broader than the simple activity of buying and selling services or goods electronically [24].

Turban et al. [25] state that e-commerce offers benefits for organizations, individual customers, and society, such as the possibility to compare prices and product on different websites; flexibility of access timewise; greater access to a wider range of products and suppliers; strengthening of competition, which can result in price reduction, generating better business opportunities for consumers; decrease in the costs of organizations; expansion of the market in which companies operate, given the possibility of access at any time, and from anywhere; obtaining assessments and recommendations through interaction and social engagement; and a decrease in the digital divide, by allowing residents of rural areas and/or developing countries to use more services and purchase what they really like. Therefore, easiness and convenience are among the key characteristics of e-commerce, given that it allows consumers to purchase a given good/service without necessarily leaving their homes [26]. In the tourist context, Carvalho et al. [27] (p. 1928) affirm that “tourism is one of the most benefited sectors by the introduction of ICT, as it is a service, and tourism activity is linked to the intangibility factor, in which there is no possibility for those who buy the service to experience it beforehand, as they do when purchasing a tangible product”; these benefits of using ICT improve revenues, and contribute to the development of the hospitality sector.

The advent of the Internet, and the advancement of the use of new technologies, changed consumer behavior, and they have come to rely on digital media to research, choose, purchase, and evaluate products and/or services [28]. According to the research carried out by Deloitte [29], this “neo-client”, since they feel at ease in the online environment, tends to do the largest number of activities online, from making new friends to purchases, given that, on the network, shopping and consumption experiences can happen in a different, more sociable way. According to Carvalho et al. [27], web consumers are those who use the Internet to buy and obtain information provided by companies or other consumers, which helps them make a purchase decision. It is clear, therefore, that the behavior of this new consumer has come far from the traditional consumer, due to their visible technological dependence.

The integration between the Internet, tourism, and ICTs associated with e-commerce influences the behavior change of the tourism consumer, because, according to Churchill and Peter [30], social influence plays an important and decisive role, due to the intangibility of tourist products, making access to information crucial for consumers to plan and execute their trips, as well as evaluate the services used throughout the experience, thus, contributing to consumers’ choices. Indeed, Vigolo and Confente [31] agree that the adoption of ICTs, and the growth of online communities, stimulated tourists to change their consumption behavior. 

However, technological transformations, and their influences on consumption behavior in tourism, especially in accommodation services, are presented in scientific studies with specific analyses, sometimes fragmented due to the focus of research, not covering conjugated variables such as habit, hedonic motivation, trust, performance expectations, security, privacy, satisfaction, ability, and frequency of use, which explain the relevance of the new paradigm of residents consumption of hosting services through digital channels. Arsal et al. [32] (p. 400) highlight that “what is missing from the literature is an examination of residents as sources of information for potential tourists”.

From this perspective, tourists become media “prosumers”, assuming a leading role in the new paradigm of participatory culture, promoted by technological interactivity [33]. In addition, tourism has the hotel sector as one of its main cornerstones in its production chain, since the means of accommodation chosen by the traveler for their stay in each place predominantly influences the assessment and experience lived by tourists on their trips. In this sense, and with justified expectations of increasing their revenues, Bagatini [34] states that the hotel sector directly influences and reflects a visitor’s experience.

Therefore, the accommodations have adapted, and currently offer the best prices on their platforms, which encourages consumers to make their reservations directly on their website, or through online price comparison platforms, without resorting to any intermediaries [35]. Thus, monetary factors are shown to be the most decisive, both from the point of view of the total quantification of the trip expense, and the low price as motivation to choose this type of accommodation [36].

Del Chiappa [37] identifies those travelers who use the Internet to make their accommodation reservations in various ways, and are divided between those who just want to find information as mere spectators, customers who seek information to help in their purchase decision, and those who, in addition to conducting research, also use the platforms to make their reservations. Indeed, according to a study by Toh et al. [38], among 78% of the respondents who use the Internet to search for information, 67% make online reservations. Therefore, it appears that consumers who conduct online searches are willing to purchase hospitality services through digital means.

Therefore, and from the perspective of the sustainable development of the hospitality sector, online hotel booking is the fastest growing trend in tourist purchases. According to Bremner [39], before the COVID-19 pandemic, global online travel sales were estimated to reach 52% by 2024, and that sales made via mobile applications would account for a quarter of all travel bookings in financial terms. Also, the accommodation booking sector reached 47% of all online sales in 2019 [39].

## 3. Conceptual Model and Hypotheses

According to Tacco [40], to consolidate electronic commerce, electronic barriers must be overcome. Therefore, for this business model to succeed, the stakeholders (consumers and sellers) must adhere to ICT, and there are several studies addressing the adoption of technology by individuals.

Internationally, studies in several areas address the use and acceptance of technology: banking services [41]; radio frequency identification service [42]; and mobile payment services [43]. In Brazil, following the international trend, some studies were conducted considering the themes of consumption, technology, and tourism, such as consumption through collective purchases [44]; internet use on smartphones [45]; and the adoption of online shared hosting platforms, such as Airbnb [46]. Additionally, according to Vera [44], the conceptual model unified theory of acceptance and use of technology 2 (UTAUT 2) is the most recent model of technology adoption, and encompasses more constructs than previous models [44].

UTAUT2 is an extension and adaptation of the unified technology acceptance and use model (UTAUT). This, in turn, stems from a review by Venkatesh et al. [47] of previous models dealing with the acceptance and use of technology, namely: the theory of reasoned action (TRA), the technology acceptance model (TAM and TAM 2), the motivational model (MM), the theory of planned behavior (TPB), the combined TAM and TPB (C-TAM-168 TPB), the model of personal computer utilization (MPCU), the innovation diffusion theory (IDT), and the social cognitive theory (SCT).

Vera [44] states that while UTAUT was developed to study the acceptance and use of technology in the context of work/organizations, UTAUT2 was adapted to the context of consumer use, and it is, therefore, more appropriate to the object of this research. The author also states that the constructs “Performance Expectancy”, “Effort Expectancy”, “Social Influence”, and “Facilitating Conditions” present in UTAUT were adapted to the context of consumption. In addition, “Hedonic Motivation”, “Price Value”, and “Habit” were included as constructs that directly influence technology acceptance and use behavior.

In summary, UTAUT2 points out that its constructs are predictors of behavioral intention, and the latter is a determining factor of use behavior. Moreover, the model foresees the constructs facilitating conditions, and habits directly influencing the use behavior. Finally, moderating variables such as age, gender, and experience continue to be used as variables of individual differences.

Therefore, for the conceptual research model shown in Figure 1, the UTAUT2 constructs of social influence and price were adopted, along with perceived risk and trust, which are variables observed in the study of Slade et al. [43]. Also, the moderators age and experience from UTAUT2 were incorporated, and the levels of income and education were added to outline the consumers’ profile, given that this survey aims to trace the relationship between the socio-demographic profile and the use of electronic commerce for purchasing accommodation services. In the words of Morgado [48], “the type of use and the motivations for using the Internet have also been shown to be a good explanation for online shopping behavior”. Therefore, the use of the Internet is related to the UTAUT2 habit construct. Finally, the construct behavioral intention was removed from this study, as it aims to analyze the factors influencing the use of electronic commerce for purchasing accommodation services, not the intention to use it. 

The constructs were built with scientific support from three authors, as described in Table 2 (adapted from Mogado, Slade et al., and Venkatesh et al. [43,48,49]), and were then analyzed by calculating the correlation of each construct with its respective variables. Also, the influence of each construct in the contracting of accommodation services was analyzed in isolation, according to the formulated hypotheses. 

As observed in the proposed research model, the constructs mentioned above directly impact the use of electronic commerce for purchasing accommodation services. 

Therefore, based on the UTAUT2 model, the following hypothesis was formulated:

**H1.** *Social influence, perceived risk, price, trust, consumer’s profile, and use of the Internet impact the use of e-commerce for purchasing accommodation services by tourists living in Fortaleza, Ceará, Brazil*. 

To identify the factors influencing tourists living in Fortaleza who use the e-commerce environment to purchase hotel and hospitality products and services, this study focused on the main hypothesis (H1), and the hypotheses derived from H1. These are based on the literature reviewed in this article and, more deeply, on the studies on costs by Ge et al. [50], season by Liu et al. [51], social relationships by He et al. [52], trust by Buhalis et al. [53], and consumer profiles by Silva and Napiri [54].

As a result of hypothesis H1, our main research hypothesis, the following secondary hypotheses were formulated to give greater scope and robustness to the central hypothesis H1. Many studies rely on demographic data to determine why consumers shop online. The study of the profile is important, as, according to Punj and Richa [55,56], consumers’ demographic variables affect their shopping preferences. Sultan [57] states that the preference for internet services is positively associated with income, home size, and propensity to innovate, and negatively with the consumer’s age. According to Lubis [58], the higher the income, the greater the chance of opting for online shopping, although there seems to be no significant difference in terms of age and education level. However, Punj [55] points out that the consumer’s age is an important factor in the preference for online shopping. 

Our first secondary hypothesis,

**H1a.** *Concerns the socio-demographic profile, represented by the variables age, education level, and income, which positively impact the use of electronic commerce for purchasing accommodation services*. 

The Internet and the development of new information technologies allow information to travel at a remarkably high speed, and be widely available. According to Morgado [48], the type and motivations for using the Internet are good explainers of online shopping behavior. Still, according to Carvalho et al. [27], it is crucial to understand the habit of using the internet, as well as how much the individuals who “inhabit” the virtual world usually consume in the online environment. Demonstrating that age appears as a considerable moderator in online shopping, Cadavez [59] reminds us that Generation Y or Millennials are more mobile, and more dependent on new technologies. Therefore, the following hypothesis was postulated:

**H1b.** 
*The use of the Internet, represented by the variables’ frequency of use, means of access, and attitude, positively impacts the use of electronic commerce for purchasing accommodation services. Moura et al. [20] state that social influence in the consumer context refers to the degree of recognition that people from the same social environment give to the importance of using technology. Therefore, the weight of the opinion of people that the consumer regards as important was considered in other studies addressing the adoption of mobile shopping services [60,61], and the use of online banking services [62]. Social influence is measured herein by how respondents view the opinions of people who are important to them, in the context of using e-commerce in making online hotel reservations. Therefore, the following hypothesis is proposed:*


**H1c.** 
*Social influence positively impacts the use of electronic commerce for purchasing accommodation services.*


The perceived risk is a common extension of UTAUT [63]. Unlike other constructs of the UTAUT model, this construct represents a complicating element in the adoption process. Indeed, the findings by Thakur and Srivastava [64] support the hypothesis that the perceived risk negatively affects the intention to use the technology. Along these lines, the following hypothesis was postulated:

**H1d.** 
*Perceived risk negatively impacts the use of e-commerce for purchasing accommodation services.*


The price value is defined as the consumer’s cognitive trade-off between the product or service’s perceived benefits and the cost to purchase them [46]. For Alalwan et al. [65], consumers cleverly assess the utilities included in the use of new systems from the perspective of the financial cost required to use a given system. Therefore, value is evaluated herein in terms of the presence or absence of an advantage concerning the prices charged to use online and digital means. Therefore:

**H1e.** 
*Price positively impacts the use of e-commerce for purchasing accommodation services.*


Lu et al., and Zhou [66,67] define trust as the subjective belief that a party will fulfill its obligations. It plays a key role in electronic financial transactions, where users are vulnerable to risks of uncertainty, and experience a feeling of loss of control. According to Moura et al. [20], the choices related to internet tourism involve online transactions, which, in turn, requires trust between the parties involved. Therefore, the following hypothesis was postulated:

**H1f.** 
*Trust positively impacts the use of e-commerce for purchasing accommodation services.*


## 4. Methodology

### 4.1. Methods Used 

This study is qualitative, with an exploratory–descriptive character, and operationalized through an online survey shared through the social networks WhatsApp and Instagram, which proved to be better suited to the “snowball” research model [68,69]. For Dewes [70], snowball sampling is indicated when there is difficulty in identifying all members of the investigated population. The social networks chosen as a tool for distributing the questionnaire are justified in this research, as they bring the ease of access and forward convenience to the respondents. However, a pre-test was conducted through a personal interview with ten respondents residing in Fortaleza, Ceará, Brazil, from November 1 to 7 November 2019. The goal was to check for flaws and difficulties in understanding the questions, and correct them before applying the questionnaire. The pre-test shows that the average time needed to answer the questionnaire is five minutes. Moreover, since the respondents did not report any difficulty or doubt regarding the questionnaire response, no adjustments to the survey were necessary.

The survey is structured in three sections, with 11 questions. Six questions have a single answer, one has a single or combined answer, one has an open answer, and three have alternatives on a 5-point Likert scale (in which 1 means “I completely disagree” and 5 means “I completely agree”). The great advantage of the Likert scale is its simplicity, as respondents can easily show their degree of agreement with any statement. Additionally, the confirmation of psychometric consistency in the metrics using this scale contributes positively to its application in the most diverse surveys [71]. Section 1 deals with information regarding the respondent’s socio-demographic profile. Five questions were used for this purpose, of which only one is open and refers to the neighborhood where the respondent resides. Furthermore, the questions present a single option, and deal with the classification of the individual in terms of age, educational level, and income, as well as whether they reside in Fortaleza, Ceará. Section 2 aims to characterize the frequency of internet use, the means of access to websites, the reasons for and the services that contribute to using it. To this end, five questions were prepared, most of which are closed questions with single answers. The questions addressing the reasons for using the internet, and what one likes to participate in in the virtual environment are closed, with answers on a 5-point Likert scale, in which 1 means “I completely disagree” and 5 means “I completely agree”. Finally, Section 3 aims to analyze the factors impacting the use of electronic commerce for purchasing accommodation services. For that, questions were prepared with closed questions, and answers on a 5-point Likert scale, where 1 means “completely disagree” and 5 means “completely agree”. The questionnaire aims to verify how the constructs social influence, perceived risk, price, and trust impact the use of e-commerce in this type of purchase.

According to IBGE [72], the city of Fortaleza has a total population of 2,686,612, among which 72.1% have access to the internet. Furthermore, in relation to the gross domestic product, Fortaleza is the largest city in the north–east region of Brazil. At the national level, the capital of the state of Ceará is among the ten largest municipalities in the country [73]. Knowing that purchasing power is directly related to the ability to travel, this fact becomes relevant, as it demonstrates that the population of Ceará has a certain purchasing power that allows them to participate in tourism. Also, according to research published by the IBGE, the north–east is ranked as the second highest region in terms of sending and receiving the most travelers from Brazil [15]. To simplify the operationalization of the survey, a non-probabilistic convenience sample was adopted, and the equation used defined that a minimum of 384 respondents was necessary. However, the success rate was 54.42%, influenced by the precariousness of the public infrastructure, exemplified by the intermittent distribution of electricity.

Sample Equation
$$N=\frac{N\cdot Z^{2}\cdot p\cdot \left(1-p\;\right)}{Z^{2}\cdot p\cdot \left(1-p\right)+e^{2}\left(N-1\right)},$$
where:

*n* = calculated sample size;

*N* = population;

*Z* = confidence interval for standard normal distribution;

*E* = margin of error;

*p* = true probability.

Source: adapted from Santos [74].

The special distribution of the sample, considering the random characteristic, corroborated the socio-economic data of the city of Fortaleza, which demonstrates the ease of access to the internet among residents of high-income neighborhoods (or neighborhoods with the highest GDP in Fortaleza).

Once this information was collected, the questionnaire was built on the Google Forms platform, and made available from 10 November 2019 to 31 January 2020. The research sample consisted of a total of 209 respondents, among which 195 were valid, as 14 respondents were excluded for not residing in Fortaleza. The sample’s margin of error is 7%, with a 95% confidence level. It should be noted that studies addressing the consumer’s profile, from Morás, Varella et al., and Tozi et al. [75,76,77], reveal a percentage of margin of error similar to, or higher than, the percentage considered in this study. It is also noteworthy that, according to Hair Júnior et. al. [78], as a general rule, a minimum of five observations per estimated parameter must be used; however, it is recommended that this coefficient reach 10 respondents for each variable. Thus, it is considered that a total of 195 valid questionnaires is sufficient for a questionnaire containing 18 variables when power is considered.

The SPSS version 22.0 software package was used for the statistical treatment of the data, by adopting a 5% significance level for statistical analysis. To describe the population and the variables analyzed in the study, descriptive statistics were used through univariate statistics. Regarding the analysis of the constructs, multivariate analysis was applied using Fisher’s exact test, factor analysis, and multiple linear regression. All the parameters in an SEM model are regression (related) coefficients. SEM is a way to perform regression; it is not an alternative method of analysis. There are different stats packages that can perform regression, but they carry out the same process. Furthermore, in this work, the regression analysis was used. Regression was used to analyze the joint influence of explanatory variables for a single response variable, with a lower number of assumptions.

In this sense, to investigate the relationship between the socio-demographic profile and the acquisition of accommodation services by electronic means, Fisher’s exact test was applied, for at least 25% of the cell values present an expected frequency lower than 5. To verify whether the items of the constructs (social influence, perceived risk, trust, and price) are strongly associated and represent a single concept, exploratory factor analysis (EFA) was used, according to principal component analysis (PCA), with varimax rotation. To demonstrate the dimensionality of the indicators, the concepts of factor loadings, Kaiser–Meyer–Olkin (KMO) and Bartlett’s tests, and explained variance were used; Cronbach’s alpha was used to test reliability

Finally, multiple linear regression was used to answer the hypotheses regarding the relationship between the variables internet use, social influence, perceived risk, trust, price, and the use of electronic commerce for purchasing accommodation services. To this end, the following assumptions are observed: normality, homoscedasticity, absence of serial autocorrelation, and multicollinearity. 

### 4.2. Data Analysis Procedures 

#### 4.2.1. Socio-Demographic Profile

The components considered for the socio-demographic profile variable are age, educational level, and salary range. Therefore, the level of significance for the age group component is 0.001. For education level and salary range, the levels of significance obtained are 0.101 and 0.044, respectively. In relation to salary range (income), Carvalho [25] recalls that this variable is commonly associated with time in such a way that the higher the income, the less time available for the individual to conduct research and make purchases outside the virtual world. Thus, the use of websites makes life easier for these people, who can shop without leaving home, or even at work. Therefore, it is possible to observe the presence of statistically significant differences in the relationship between age and salary ranges and the use of electronic commerce for purchasing accommodation services (α < 0.05). This, in turn, verifies the positive impact of the components of the studied variables and the purchase of accommodation services through digital means. This refutes hypothesis Ha, as there is no significant relationship between the level of education and the use of e-commerce for online bookings.

As to the use of the Internet, Morgado [48] states that its type and motivations are presented as good explaining factors for the online shopping behavior, both regarding the objective aspects (frequency), and the motivational aspects of such use (attitude). In turn, regarding the means of access, Sousa [79] states that online reservations are one of the major trends to emerge from new technologies, and that this service, made available through mobile applications, is likely to become the primary distribution channel of tourism. Therefore, an attempt was made to evaluate the relationship between electronic commerce and the purchase of accommodation services. In this sense, the relationship between the use of the Internet and the use of electronic commerce for making online reservations is analyzed.

#### 4.2.2. Respondents’ Profile

Table 3 and Figure 1 shows the respondents’ socio-demographic profiles. The age range of this sample is between 19 and over 60 years of age, with the age group between 31 and 40 years old being the most represented, with a percentage of 34.36%. As for the level of education, 40.51% have a postgraduate degree, followed by 23.59% who have a complete higher education. The research shows that the interviewees have a high level of education, which is consistent with the social study of Fortaleza’s population since, in 2017, the municipal educational human development index was 0.767, on a scale of 0 to 1, being the third-best index in relation to the other regions concentrated in the north–east [80]. Regarding income, two groups have greater representation, with 26.67% of the sample earning between 5 and 10 minimum wages, followed by those earning more than 20 minimum wages, 23.08%, showing the high economic level of the interviewees.

As for the frequency of internet use, the results show that 96.92% of respondents access the Internet daily. Interestingly, no respondent chose the option “Rarely”, which proves that daily access to the Internet is part of the respondents’ routine. These results are similar to data released by Cetic.br, which reports that 90% of all Brazilians access the Internet every day, 7% at least once a week, and 2% at least once a month [81].

#### 4.2.3. Means Used to Access the Internet 

Table 4 shows the means used to access the Internet. It appears that 86.14% of respondents use their mobile phones as a means to access and browse the Internet, both through apps and mobile versions of websites. This result corroborates data presented by Poggi [82], who finds that 87% of digital moments happen on mobile devices. This is probably also a fact that the research has the internet as a collection instrument, through digital media; thus, it may be biased. The research being distributed through social media implies that when buying, which is our focus, consumers use their mobile device to do so. The numbers demonstrate that people prefer smartphones both to access the website in the mobile version, and to use the application itself.

#### 4.2.4. Respondents’ Behavior on Internet 

To identify users’ behavior on the Internet, respondents were presented with options for activities, namely: the use of the Internet to access bank information and pay bills; the desire to buy products/services online, as well as to experiment with new technologies; and the habit of comparing prices over the Internet. This study concludes that access to internet banking services, both to access bank information and pay bills, totals 72.82% and 74.35% of usage, respectively. This indicates that its use is disseminated among the respondents, corroborating the data released by the Brazilian Federation of Banks (FEBRBAN), which show that the number of bank transactions through digital means grows every year and, in 2019, reached 56.2 billion transactions. Indeed, this value represents 62.5% of all bank transactions carried out in 2019 [83]. Next is the preference for using the Internet to compare prices and try new technologies, with 69.22% and 57.95%, respectively. Finally, totaling 54.87%, we find that users use the Internet to purchase products and services. However, according to the newsletter Global Online Retail Spending, the number of internet consumers increases every year [84].

Still, concerning these items, the taste for shopping for products and services online presents an 18.39 standard deviation concerning the level of agreement, with the lowest deviation index. In turn, the use of the Internet to pay bills shows the highest standard deviation index among the levels of agreement, at 27.32.

## 5. Results and Discussion

Table 5 shows the perceptions of the residents of Fortaleza, Ceará, regarding the constructs social influence, trust, price, and perceived risk in the context of accommodation services. Regarding the social influence construct, item SI2 (The people who I live with often use the internet to purchase accommodation services) has the highest mean, the highest percentage of agreement, as well as the lowest standard deviation, which represents the level dispersion of the data. Also, more than 71% of Fortaleza residents perceive social influence as related to the consumption of accommodation services on the internet.

Regarding the construct price, the item P1 (The prices offered on the Internet are reasonable for the purchase of accommodation services) has a mean of 3.91 on the numerical scale, and a standard deviation of 1.031. Indeed, 79.17% of the respondents agree with this statement, 7.81% claim to be indifferent, and the remaining 13.02% disagree. In other words, price is truly relevant for the acquisition of accommodation services by electronic means, with over 75%of participants agreeing with the correlated statements.

As for security associated with perceived risk, the item PR3 (I feel insecure by sending sensitive information over the Internet to purchase accommodation services) reaches a 3.22 mean value for the scale, and a standard deviation of 1.234. A total of 50.52% of respondents agree with the statement, 13.02% are indifferent, and 36.46% say they do not agree. 

Finally, regarding the trust construct, the item T1 (I trust the information and transactions carried out through the Internet for the acquisition of accommodation services) is the construct with the best indexes, reaching a mean value of 3.59 on the scale, and a standard deviation of 1.138. A total of 67.88% of the respondents claim to agree with the respective statement, 8.29% claim to neither agree nor disagree, whereas 23.83% disagree. 

It is our conviction that these results are in line with the results of the reviewed studies, namely Moura et al. [18], Thakur and Srivastava [61], Alalwan et al. [62], Lu et al. [63], and Zhou [64].

The analysis shows that if we use the mean of the dependent variable purchase of accommodation services, the total sum of squares is 302,533. Using the consumer profile independent variable, this residue drops to 270,838. In addition, the significance at the 0.000 level demonstrates the existence of a significant relationship between the dependent variable (use of e-commerce) and the set of independent variables (the model constructs). Regarding the assumptions, the normality of the residuals is confirmed by the Kolmogorov–Smirnov (K–S) test, in which the level of significance must be greater than 0.05. Indeed, this study obtains a value of 0.255. Regarding the absence of serial autocorrelation, the index is 1.781. For a sample with 195 items, with three independent variables, the Durbin–Watson table defines the non-conclusive interval as the critical values below dL = 1.643 and above dU = 1.704.

The assumption of multicollinearity between the independent variables is examined by the variance inflation factor (VIF) index. If VIF = 1, there is no multicollinearity; if 1 > VIF < 10, there is acceptable multicollinearity; and if VIF > 10, multicollinearity is problematic [85]. Therefore, according to the Table 6, it appears that such assumption is not violated, given that the indices found are in the range of acceptable multicollinearity.

Finally, homoscedasticity reaches 0.209. According to Hair Júnior [86], for this assumption to be met, the level of significance must be greater than 0.05. Shows that the variable “Attitude” is the variable that best explains the dependent variable use of the Internet for purchasing accommodation services, as it is the only one with a significance level below 0.05. The variables “Frequency” and “Means of Access” are not significant in terms of the use of the Internet for purchasing accommodation services, given that the level of significance is greater than 0.05. This, therefore, refutes hypothesis Hb. These contradict the results of the studies of Morgado [46], and Carvalho et al. [25].

As this study aims to measure concepts, it is important to assess the dimensionality and reliability of these indicators. According to Vera [44], the dimensionality analysis aims to ascertain whether the indicators are strongly associated with each other, thus, corresponding to a single concept. In this sense, the latent variables of the research are grouped into four concepts: social influence, price, perceived risk, and trust, and the one dimensionality analysis of each of these constructs is verified through factor analysis using the KMO values, the significance of Bartlett’s test for sphericity, and explained variance. Reliability is measured by calculating Cronbach’s alpha. Vera [44] points out that the value assumed by alpha is between 0 and 1, and the closer to 1, the greater the reliability of the construct dimensions. Cronbach’s alpha is already inserted in the Table 7, where we analyze the dimensionality and the relationship of the constructs. 

According to the table above, it appears that, for each construct, a single component is extracted. The items’ factor loads are high, all above 0.50, as recommended by Corrar et al. [85]. The KMO has an adequate value according to the literature, that is, above 0.50, indicating the degree of data explanation from the factors found in factor analysis. Furthermore, the significance of Bartlett’s test of sphericity shows that correlations between variables are sufficient. It should also be noted that the total explained variance of the constructs is above the 60% value proposed by Corrar et al. [85]. Additionally, as to the reliability, the Cronbach’s alpha of all constructs exceeds the 0.70 value recommended in the literature.

In the analysis of the relationships between the independent variables social influence, price, perceived risk, and trust, and the dependent variable use of e-commerce for purchasing accommodation services, multiple linear regression is applied.

Initially, all constructs are simultaneously related to the dependent variable use of electronic commerce to make online reservations. Corrar et al. [85] state that an F-ANOVA test with a significance less than 0.05 indicates that the statistical variable influences the dependent variable, and the model is, therefore, significant. Below is the model summary with the correlation coefficient (R), the coefficient of determination (R square), the adjusted coefficient of determination (adjusted R square), and the standard error of the estimate of the multiple linear regression of the independent variables, in relation to the dependent variable the use of electronic commerce for making online reservations.

That said, the existence of a significant relationship between the dependent variable and the set of explanatory variables is verified, given that the F-ANOVA statistic points to a significance level of 0.000. Furthermore, using the average of the dependent variable, purchase of accommodation services, the total sum of squares is 302,533. Using the independent variables, this residual drops to 2236,565.

Regarding the assumptions that guarantee the tests’ integrity, and the model’s significance, the normality of the residuals reach a significance level of 0.119, that is, higher than what is proposed in the literature. The second considered assumption is the absence of serial autocorrelation, with a value of 2.004, according to the Durbin–Watson statistic. According to Corrar et al. [85], DW statistic values close to 2 meet the assumption, thus, verifying the validity of this assumption. The third analyzed assumption refers to multicollinearity between the independent variables, examined through the variance inflation factor (VIF) index. As can be seen in Table 8 and Table 9, the indexes remain within the range of acceptable multicollinearity, according to the literature. Finally, homoscedasticity is verified, indicating the non-violation of this assumption, given that the level of significance totals 0.470.

Once the assumptions are confirmed, we proceed to analyze each construct with the dependent variable use of electronic commerce for purchasing accommodation services. According to the table above, the following levels of significance are observed for each independent variable, related to the dependent variable. It can be noted that the variables social influence, perceived risk, and price are the variables that best explain the dependent variable use of e-commerce for purchasing accommodation services, as the levels of significance are all below 0.05. It is worth mentioning that the perceived risk construct shows negative variation, indicating that this variable decreases the probability of using e-commerce to make online reservations.

Given the above, the hypotheses Hc, Hd, and He are confirmed through multiple linear regression, while Hf is refuted. The final model of the study is presented in Figure 2, where the dotted line between the variables represents the absence of a significant relationship.

## 6. Conclusions and Future Work

This study sought to identify what factors impact the purchase of accommodation services by the citizens of the municipality of Fortaleza, capital of the Brazilian state of Ceará (CE), leading to the increase in hotel revenues. This is a significant contribution to the literature, considering that there is a gap in investigations regarding the use of information and communication technologies, making use of the residents of a tourist destination as a source of information, in the expectation of being potential tourists, for the acquisition and reservation of accommodation services. In this way, the profile of the respondents is identified through data analysis, and shows that most respondents (34.36%) are 31 to 40 years old, have a graduate degree, and an income ranging from 5 to 10 minimum wages per month (for sampling purposes, a value of the minimum monthly wage in Brazil of approximately USD 205 was considered).

The daily frequency of Internet use reaches 96.92%, showing that internet access is a key part of the respondents’ routine. Furthermore, the mobile phone proves to be the most-used device to access the Internet (48.71%), through mobile applications.

To measure the factors impacting the purchase of accommodation services via e-commerce, the variables price, social influence, perceived risk, trust, consumer’s profile, and use of the Internet were outlined, all of which are present in the hypotheses formulated for the main objective. Therefore, the analysis of the data reveals that, in the opposite direction to the educational level, the variables age and income influence the use of electronic commerce to purchase hosting services, allowing the confirmation of the initial hypothesis (Ha). This study, by seeking to contribute to the expansion of the hotel sector’s capabilities in market segmentation, and, thus, increase its revenues and identify potential consumers more easily, is innovative in the variables studied, among which are some not usually used by segmentation methodologies adopted by companies, such as social influence and risk perception.

Likewise, the relationship between the variables social influence, perceived risk, and price is measured with the variable use of e-commerce for purchasing accommodation services. As anticipated by Kim and Srivastava, Tzavlopoulos et al., and Jo et al. [87,88,89], who recognize the influence of the three variables in the use of e-commerce, the analysis confirms the hypotheses Hc, Hd, and He, while Hf is refuted. Therefore, it is up to companies to seek to improve trust, through initiatives that may result in a better relationship with consumers, such as social proof, displaying high-resolution images, and resorting to social, psychological, and economic instincts, etc., once this hypothesis is refuted.

The practical implication of the study is the contribution to the sustainable growth of hospitality companies in the context of modern management which, when considering the use of the Internet as a business element, should focus on maximizing the positive aspects related to the factors that influence the use of e-commerce in the purchase of accommodation services, and minimizing the aspects with a negative impact. As theoretical contribution, the greatest impact of the study findings is clearly the change to the conceptual research model initially presented (Figure 1), based on the literature review, to a revised conceptual research model (Figure 2), depending on the confirmation or rejection of the formulated hypotheses. To broaden the discussion on the topic, it is suggested that future studies rely on larger samples, and address other factors regarding the purchase of accommodation services through e-commerce. Another suggestion is to further investigate the tourist consumer behavior; the factors influencing the non-use of e-commerce can be investigated, or the scope of the research can be expanded to other regions, to compare the consumer behavior of Brazilian tourists. In this sense, for the development of the hotel sector facing digital challenges, this study shows that companies must increasingly improve the offer of online reservation services through mobile devices, as the population analyzed seems to favor this means of access.

It should be noted that over the course of this study, the COVID-19 pandemic broke out all over the world. Several sectors of the economy were affected, but the tourism sector is among the most affected ones. According to Occhialini [90], in 2020 alone, 100 million workers lost their jobs, along with losses accounting for USD 2.7 trillion in the world GDP in this sector alone. Furthermore, the World Travel and Tourism Council is betting on passenger tracking via applications, less cash circulation, and improved hygiene [90]. Therefore, the need to continue advancing in the use of technology to improve the behavior of organizations is evident

Regarding the limitations of our research, firstly, a limitation can be attributed to the way our questionnaire was distributed, via WhatsApp. But we know that, in Brazil, this application is one of the most used, because, according to Purz [69], there are 118.5 million users in Brazil, which makes us believe that the medium is interesting, but the limitation may be due to contacts, considering the adequacy of the research to the “snowball” research model. Therefore, even with Table 1, showing the demographic importance of Fortaleza in Brazil, we cannot generalize, due to the sampling bias. In addition, another limitation that we can point out is the size of the sample obtained, which, while sufficient for the research in question, may have limited the results to only a part of reality, not being able to generalize as a whole. In addition, another limitation may be that other factors also impact the use of e-commerce in online reservations, which were not addressed in our research.

Therefore, it is suggested that future works expand and diversify the data collection instruments, as well the sample used in the research.

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
