# Peer review of "The Development of the Hospitality Sector Facing the Digital Challenge"

_behavsci, 2022, doi:10.3390/bs12060192_

Round 1

Reviewer 1 Report

First, I applaud the authors for an interesting study. However, I have concerns which are listed below:

  1. There is considerable literature in information search that finds that age, income, education and time pressure (Ratchford et al. 2003; Singh et al 2014; Singh & Jang 2022) influence information search, product search in multi channel environments. So it is not clear why hypothesis related to age, income offer new insights. The authors need to strengthen their hypotheses in light of current research.
  2. Methodologically, authors use EFA and then multiple linear regression to test their hypotheses. The conceptual framework aligns with structural equation modeling (SEM). This methodology has been used in most recent studies employing the TAM framework. My humble suggestion is that authors justify their methodology.

Author Response

Authors’ Response to Reviewers’ Comments

Manuscript title:

Title: The development of the hospitality sector facing the digital challenge

Manuscript ID: Manuscript ID: behavsci-1704278

Dear Respected Reviewers,

We would like to thank you for acknowledging the merit and value of our paper, as well as for your valuable contributions. We have carefully considered, your comments and suggestions, which allowed us to significantly improve our manuscript.  Please find our response to each individual comment/suggestion on the table below. In line with each of your comments/suggestions, there are some specific revisions to the text as highlighted in red within the revised manuscript.

Many thanks for your time and effort.

Kind regards,

Reviewer 1:

Comment

Response

First, I applaud the authors for an interesting study.

Many thanks for your interesting and acknowledging the merit of our paper. 

Thanks a lot for your support, your suggestions have helped to improve further this work.

There is considerable literature in information search that finds that age, income, education and time pressure (Ratchford et al. 2003; Singh et al 2014; Singh & Jang 2022) influence information search, product search in multi-channel environments. So it is not clear why hypothesis related to age, income offer new insights. The authors need to strengthen their hypotheses in light of current research

We thank you for your suggestion and we understand the hypotheses related to age and income offer new insights, given the sum of other hypotheses and other factors (conditional or not) added to online shopping and the usability of digital channels for consultations and purchases.

Methodologically, authors use EFA and then multiple linear regression to test their hypotheses. The conceptual framework aligns with structural equation modeling (SEM). This methodology has been used in most recent studies employing the TAM framework. My humble suggestion is that authors justify their methodology.

We thank again your comment. All the parameters in a SEM model are regression (related) coefficients. SEM is a way to do regression, it's not an alternative method of analysis. There are different stats packages that can do regression, but they do the same thing. Furthermore, in this work, the regression analysis was used. Regression was used to analyze the joint influence of explanatory variables for a single response variable, with a lower number of assumptions.

We consulted 2 statisticians during the research and during the writing of this paper, in addition, we forwarded your comment to them, and they gave us this support for clarification, in addition, we reread these two sources to be sure: Bollen, K. A., & Pearl, J. (2013). Eight myths about causality and structural equation modeling. In S.L. Morgan (Ed.), Handbook of Causal Analysis for Social Research (pp. 301-328). Dordrecht: Springer.

and

Chen, B., & Pearl, J. (2013). Regression and causation: A critical examination of six econometric textbooks. Real-World Economics Review, 65, 2-20.

Reviewer 2 Report

  1. The title is misleading. The study and most of the discussions are about online hotel booking, using a well-established framework of UTAUT. Using the development of the hospitality sector facing the digital challenge as a title provide high expectation for the reader, such as other hospitality services and other digital applications that has been much applied in this industry (such as: chatbots, service robots, AR/VR, AI-based checkout, etc). However, this study only discuss and limit it on e-commerce using mobile and computer which has been quite outdated
  2. The results is quite straightforward and predictable. Some variables have been heavily used in the previous studies. Therefore, I am afraid that the contributions are very limited.
  3. Sample size is too small, especially for one-country sample
  4. The hypothesis testing for Ha and Hb are less explained and not very clear. This part should provide better explanation especially regarding to the customer profile and persona

Author Response

Authors’ Response to Reviewers’ Comments

Manuscript title:

Title: The development of the hospitality sector facing the digital challenge

Manuscript ID: Manuscript ID: behavsci-1704278

Dear Respected Reviewers,

We would like to thank you for acknowledging the merit and value of our paper, as well as for your valuable contributions. We have carefully considered, your comments and suggestions, which allowed us to significantly improve our manuscript.  Please find our response to each individual comment/suggestion on the table below. In line with each of your comments/suggestions, there are some specific revisions to the text as highlighted in red within the revised manuscript.

Many thanks for your time and effort.

Kind regards,

Reviewer 2:

Comment

Response

The title is misleading. The study and most of the discussions are about online hotel booking, using a well-established framework of UTAUT. Using the development of the hospitality sector facing the digital challenge as a title provide high expectation for the reader, such as other hospitality services and other digital applications that has been much applied in this industry (such as: chatbots, service robots, AR/VR, AI-based checkout, etc). However, this study only discuss and limit it on e-commerce using mobile and computer which has been quite outdated.

With all due respect, the authors express their disagreement.

Another 5 reviewers-evaluators did not find this same view, even because the digital challenge comes from knowing if people who consume hospitality and tourism actually use digital (how is their relationship with digital? How do they understand digital? fear of using? among other hypotheses that UTAUT2 itself, which is a rigorous and already established procedure, deals with the usability of digital and technology), therefore, dealing with more technical issues of technology such as chatbots, robots and etc. diagnosis of what we did. Therefore, at this moment, we are willing to move forward with another research, then with what you considered from our beginning of the paper. However, as we author and even the other 5 reviewers do not understand it this way, we believe that the title is within expectations and what was achieved with the results of our production.

The results is quite straightforward and predictable. Some variables have been heavily used in the previous studies. Therefore, I am afraid that the contributions are very limited.

The authors disagree as a review cannot be done on the basis of assumptions or personal impressions.

Nevertheless, two of the other reviewers agreed with the structure of the conclusions, highlighting the limitations and future lines of research presented; in addition, one of the other reviewers recommended changing the title of the section "Conclusions" to "Conclusions and Future Work", because in his opinion it is clear that they indicate several additional efforts for the future.

Sample size is too small, especially for one-country sample.

With all due respect, the authors express their disagreement.

Like argued buy reviewers 5 and 6, the authors consider although the study is carried out only among the inhabitants of a single Brazilian city, the sample is large enough to be significant.

The hypothesis testing for Ha and Hb are less explained and not very clear. This part should provide better explanation especially regarding to the customer profile and persona

H1: Social influence, perceived risk, price, trust, consumer’s profile, and use of the Internet impact the use of e-commerce for purchasing accommodation services by tourists in Fortaleza, Ceará, Brazil. This hypothesis needs to be tested with another hypothesis that contradicts it.

Ha: Social influence, perceived risk, price, trust, consumer’s profile (represented by the variables age, education, use of the Internet, income) do not positively impact the use of electronic commerce for accommodation services.

Hb: Social influence, perceived risk, price, trust, consumer's profile, and use of the Internet consumer don't positively impact the use of e-commerce for purchasing accommodation services tourists Fortaleza, Ceará, Brazil, due to the use of internet, represented by the variables' frequency of use, means of access, and attitude.

In relation to salary range (income), Carvalho [25] recalls that this variable is commonly associated with time in such a way that the higher the income, the less time available by the individual to do research and purchases outside the virtual world. Thus, the use of websites makes life easier for these people, who can shop without leaving home or even at work.

(Ha) demonstrating that age appears as a considerable moderator in online shopping. Cadavez (2017) reminds us that Generation Y or Millennials are more mobile and more dependent on new technologies.

-          Cadavez, C. (2017). “But what world is this?”, or how the dissemination of cultural enjoyment practices for millennial tourists must be different – a case study thinking about museums. Revista Iberoamericana de Turismo – RITUR (Iberoamerican Journal of Tourism), Penedo, v. 7, n. 3, p. 215-228, dez. 2017.

Reviewer 3 Report

Authors must identify the limitations imposed by the methodologic approach followed in the paper.
The discussion of the results must be improved. It is not clear how the paper fits among the other studies done in this area.
The authors’ conclusions are not appropriate given the results of their analysis.
It is not clear what is the practical and theoretical contribution of the paper. What is new?

Author Response

Authors’ Response to Reviewers’ Comments

Manuscript title:

Title: The development of the hospitality sector facing the digital challenge

Manuscript ID: Manuscript ID: behavsci-1704278

Dear Respected Reviewers,

We would like to thank you for acknowledging the merit and value of our paper, as well as for your valuable contributions. We have carefully considered, your comments and suggestions, which allowed us to significantly improve our manuscript.  Please find our response to each individual comment/suggestion on the table below. In line with each of your comments/suggestions, there are some specific revisions to the text as highlighted in red within the revised manuscript.

Many thanks for your time and effort.

Kind regards,

Reviewer 3:

Comment

Response

Authors must identify the limitations imposed by the methodologic approach followed in the paper.

In our first version, we had already included the limitations of the study in the conclusions, including opening space for future research on the subject. In this way we understand that the limitation is already on lines 700, 701, 702, 703, 704, 705, 706.

The discussion of the results must be improved. It is not clear how the paper fits among the other studies done in this area.

We thank you for your suggestion and in the revised version we improved this section (lines 525, 526, 527; and 553, 554).

The authors’ conclusions are not appropriate given the results of their analysis.

We thank you for your suggestion and in the revised version we improved this section (lines 650, 651, 652, 653, 654, 655, 656, 657; and 671, 672, 673, 674, 675, 676, 677, 678, 679).

It is not clear what is the practical and theoretical contribution of the paper. What is new?

We are grateful for the comment, nevertheless, two of the other reviewers agreed with the structure of the conclusions, highlighting the limitations and future lines of research presented; in addition, one of the other reviewers recommended changing the title of the section "Conclusions" to "Conclusions and Future Work", because in his opinion it is clear that they indicate several additional efforts for the future.

What we mentioned did not prevent the acceptance of your critical opinion, so in “chapter” 6, the practical implications for management and the theoretical implications of the study were highlighted with the revision of the model, which we consider the main innovative issue.

Reviewer 4 Report

This is an interesting contributions, but it requires a special attention in some aspects, such as:

a) Abstract:

> The sentence started with "Additionaly" let the reader think about a previous contribution which actually does not exist. A idea is stated before not a contribution;

> In Line 17 exists " these residents", which are not early mentioned;

b) Introduction:

> It is recommend a extra final paragraph with the paper structure and why for those sections;

> Fortaleza(?)/Ceara(?) and Fortaleza(?)/CE(?) appears before line 75. It is very importante to be clear what does these words means (City? State? Province?). The authors are considering a tacit knowledge from the readers.

It is required a clear explantion about these places.

c)   Refeering to the previous comment it occurs also at line 474 with the word FEBRABAN. There is no indication what this means, the reader shoud find in the reference what is the meaning.

d) Tables 7 and 8 requires attention because there are error refeering to the change of dot (.) for the comma (,) inside these tables.

e) Conclusions

> It is strong recommended the change of the sections title to "Conclusions and Future Work", because it is clear that the present work inidcates several addition effords for the future 

f) It is recommend a further read from the authors to clean local knowledge which they belive that any reader around the world could understand local words or organisations (e.g. city/state, it could be state/city) 

Author Response

Authors’ Response to Reviewers’ Comments

Manuscript title:

Title: The development of the hospitality sector facing the digital challenge

Manuscript ID: Manuscript ID: behavsci-1704278

Dear Respected Reviewers,

We would like to thank you for acknowledging the merit and value of our paper, as well as for your valuable contributions. We have carefully considered, your comments and suggestions, which allowed us to significantly improve our manuscript.  Please find our response to each individual comment/suggestion on the table below. In line with each of your comments/suggestions, there are some specific revisions to the text as highlighted in red within the revised manuscript.

Many thanks for your time and effort.

Kind regards,

Reviewer 4:

Comment

Response

This is an interesting contributions, but it requires a special attention in some aspects, such as:

a) Abstract:

> The sentence started with "Additionaly" let the reader think about a previous contribution which actually does not exist. A idea is stated before not a contribution;

> In Line 17 exists " these residents", which are not early mentioned;

b) Introduction:

> It is recommend a extra final paragraph with the paper structure and why for those sections;

> Fortaleza(?)/Ceara(?) and Fortaleza(?)/CE(?) appears before line 75. It is very importante to be clear what does these words means (City? State? Province?). The authors are considering a tacit knowledge from the readers.

It is required a clear explantion about these places.

Changes made and the locations explained for those who do not know the study area territorially. Thank you for paying attention to this and thinking of readers.

c)   Refeering to the previous comment it occurs also at line 474 with the word FEBRABAN. There is no indication what this means, the reader shoud find in the reference what is the meaning.

Changes made and explained.

d) Tables 7 and 8 requires attention because there are error refeering to the change of dot (.) for the comma (,) inside these tables.

Changes made following the pattern and replacing with commas.

e) Conclusions

> It is strong recommended the change of the sections title to "Conclusions and Future Work", because it is clear that the present work inidcates several addition effords for the future

f) It is recommend a further read from the authors to clean local knowledge which they belive that any reader around the world could understand local words or organisations (e.g. city/state, it could be state/city)

We thank you for your careful and careful reading of our paper, all your suggestions were accepted, and we appreciate the requested improvements. This has already been changed in the text as suggested.

Reviewer 5 Report

The manuscript presents a study that tries to identify the factors that have a greater impact on the "use of e-commerce when hiring hosting services".

Although the study is carried out only among the inhabitants of a single Brazilian city, the sample is large enough to be significant. The greatest limitation of the study lies in the method of selection and access to the volunteers who participate in the study. The use of whatsup as the only capture and communication mechanism can produce a bias in the results. In any case, the study can be considered relevant, since the potential population that is accessed through this social network can be considered broad and of special interest for the study addressed.

The article has sufficient quality for publication in its current state, although it is recommended to improve figure 2 (so that it is easier to differentiate the hypotheses with sufficient support from the refuted ones), and add graphics (e.g. bar charts or pie charts) with the data of some of the tables (especially table 2), in order to facilitate its reading.

Author Response

Authors’ Response to Reviewers’ Comments

Manuscript title:

Title: The development of the hospitality sector facing the digital challenge

Manuscript ID: Manuscript ID: behavsci-1704278

Dear Respected Reviewers,

We would like to thank you for acknowledging the merit and value of our paper, as well as for your valuable contributions. We have carefully considered, your comments and suggestions, which allowed us to significantly improve our manuscript.  Please find our response to each individual comment/suggestion on the table below. In line with each of your comments/suggestions, there are some specific revisions to the text as highlighted in red within the revised manuscript.

Many thanks for your time and effort.

Kind regards,

Reviewer 5:

Comment

Response

The manuscript presents a study that tries to identify the factors that have a greater impact on the "use of e-commerce when hiring hosting services".

Although the study is carried out only among the inhabitants of a single Brazilian city, the sample is large enough to be significant. The greatest limitation of the study lies in the method of selection and access to the volunteers who participate in the study. The use of whatsup as the only capture and communication mechanism can produce a bias in the results. In any case, the study can be considered relevant, since the potential population that is accessed through this social network can be considered broad and of special interest for the study addressed.

The article has sufficient quality for publication in its current state, although it is recommended to improve figure 2 (so that it is easier to differentiate the hypotheses with sufficient support from the refuted ones), and add graphics (e.g. bar charts or pie charts) with the data of some of the tables (especially table 2), in order to facilitate its reading.

Many thanks for your interesting and acknowledging the merit of our paper.

Thanks a lot for your support, your suggestions have helped to improve further this work. In the revised version we improved Figure 2, and we have inserted the charts as requested to give greater visibility to the data.

Reviewer 6 Report

I think it is a very topical article with a very interesting subject. The objectives and results are well written. In addition, the conclusions, limitations and future lines of research are included. Therefore, in my opinion this article should be accepted and published.

Author Response

Authors’ Response to Reviewers’ Comments

Manuscript title:

Title: The development of the hospitality sector facing the digital challenge

Manuscript ID: Manuscript ID: behavsci-1704278

Dear Respected Reviewers,

We would like to thank you for acknowledging the merit and value of our paper, as well as for your valuable contributions. We have carefully considered, your comments and suggestions, which allowed us to significantly improve our manuscript.  Please find our response to each individual comment/suggestion on the table below. In line with each of your comments/suggestions, there are some specific revisions to the text as highlighted in red within the revised manuscript.

Many thanks for your time and effort.

Kind regards,

Reviewer 6:

Comment

Response

I think it is a very topical article with a very interesting subject. The objectives and results are well written. In addition, the conclusions, limitations and future lines of research are included. Therefore, in my opinion this article should be accepted and published.

Many thanks for your interesting and acknowledging the merit of our paper. 

Thanks a lot for your support.

Round 2

Reviewer 2 Report

The rebuttal does not touch the main issues addressed in the previous comments

Reviewer 3 Report

I believe this is an enhanced version of the paper.
